# Loneliness and Depression among Polish High-School Students

**DOI:** 10.3390/ijerph18041706

**Published:** 2021-02-10

**Authors:** Beata Dziedzic, Paulina Sarwa, Ewa Kobos, Zofia Sienkiewicz, Anna Idzik, Mariusz Wysokiński, Wiesław Fidecki

**Affiliations:** 1Department of Nursing Development, Social and Medical Sciences, Faculty of Health Sciences, Medical University of Warsaw, Żwirki i Wigury 61, 02-091 Warsaw, Poland; ekobos@wum.edu.pl (E.K.); zofia.sienkiewicz@wum.edu.pl (Z.S.); anna.idzik@wum.edu.pl (A.I.); 2Central Teaching Hospital of the Ministry of the Interior and Administration, Wołoska 137, 02-507 Warszawa, Poland; sarwapaulina96@gmail.com; 3Department of Basic Nursing and Medical Teaching, Chair of Development in Nursing, Faculty of Health Sciences, Medical University of Lublin, 20-081 Lublin, Poland; mariusz.wysokinski@umlub.pl (M.W.); wieslaw.fidecki@umlub.pl (W.F.)

**Keywords:** anxiety, depression, aggression, loneliness, teenagers

## Abstract

Introduction: Having impaired relations and limited interpersonal contact is associated with a sense of loneliness, and can result in a number of mental disorders, including the development of depression. Approximately one in five adolescents in the world suffers from depression, and first episodes of such are occurring at increasingly young ages. Due to a lack of appropriate support from parents, teachers and the healthcare system, the young person feels alone when dealing with their problem. Aims: The aims of this study are to determine the prevalence of anxiety, depression, aggression and sense of loneliness among high school students, and to analyze a correlation between loneliness and depression. Materials and methods: The study was conducted on 300 high school students in Poland. The study material was collected using the Hospital Anxiety and Depression Scale (HADS-M) and De Jong Gierveld Loneliness Scale (DJGLS). Results: A feeling of loneliness correlated significantly with depressive disorders (*p* < 0.005), with the strongest effect between the total HADS-M score and the total loneliness scale score (*r* = 0.61). The overall presence of disorders as per HADS-M was found to be 23%, and borderline conditions were found in 19.3%. In 24% of the students, disorders were revealed on the anxiety subscale and in 46.3% on the aggression subscale. On DJGLS, a very severe sense of loneliness was observed in 6.67% of the subjects, and in 42.3% of them, a moderate feeling of loneliness was indicated. On the social loneliness subscale, a severe sense of loneliness was found in 22.7%, while on the emotional loneliness subscale, it was found in 16.7% of the subjects. Conclusions: In this study, a quarter of the student participants experienced anxiety and depression disorders. Students showing higher levels of anxiety, depression, and aggression also showed enhanced loneliness. Girls showed higher levels of anxiety, depression and aggression, as well as emotional loneliness.

## 1. Introduction

According to the World Health Organization (WHO), adolescents account for 1.2 billion of the world’s population [1], and 20% of them suffer from mental health problems, with anxiety and depression being the most common [2]. Every year, 1.1 million young people die. The second most common cause of death in this age group is suicide and interpersonal violence. Undoubtedly, the occurrence of mental disorders and the associated mortality depends on age, gender and geography [1]. In addition, a number of factors that reduce the possibility of receiving quick help can cause such disorders to develop [1] and suicide rates to rise, the predictors of which can be depressive disorders [3,4].

Herpertz-Dahlmann et al. showed that mental disorders such as depression, social anxiety, and eating disorders are more common in girls. Their prevalence ranges from 12% to 23%, depending on the diagnostic instruments used. Disruptive disorders, e.g., disorders of social behavior, are more common in boys, with a worldwide prevalence of approximately 5% to 10% [5]. Bor et al. showed that one in five adolescents experienced mental health problems, again with a higher prevalence among girls [6].

The 2014 data for the European Union showed that symptoms of depression were observed in 4.8% of people in the age group of 15–19 years. The highest rates were reported for Iceland, Germany, Sweden, Ireland, Denmark, Luxembourg and Slovenia, respectively (13.9%, 11%, 10.6%, 10.1%, 10%, 9.2%, 7.5%). In Poland, the percentage of adolescents with depressive disorders was 2.2% [7]. According to an analysis conducted by the Central Statistical Office of 5407 15–19-year-olds in Poland in 1996, one in ten experienced mental suffering [8]. Woynarowska et al. showed in their study in Poland that 25.2% of 15–16-year-olds exceeded the thresholds of anxiety and depression justifying specialist consultation [9].

As indicated in the literature, mental health problems that begin at an early age may recur in adulthood [10]. The symptoms of childhood-onset anxiety and depression were also shown to continue later in life [11]. Some studies have shown that the incidence of depressive disorders increases from 4% at the age of 15 years to 16% at the age of 18 years [12]. Another study estimated an increase from 8.7% at the age of 13–14 years to 15.7% at the age of 17–18 years [13].

In a study evaluating the worldwide prevalence of depressive disorders, their low diagnosis rate was pointed out [14]. Early detection of the problem and the use of appropriate interventions are necessary to reduce its scale. Untreated abnormal states cause developmental disorders to deteriorate in young people, limiting their educational and life potential [15]. Progressing mental health problems have a negative impact on family relations, contact with friends and participation in social life. According to the WHO, young people with mental health problems are stigmatized in society and overlooked by healthcare systems [16]. For this reason, WHO strategies for its member states call for increased implementation of active measures for mental health; the theme of the 2018 World Mental Health Day was “Young People and Mental Health in the Changing World” [17].

Loneliness is a negative feeling that arises when a person’s social needs are unmet by the quantity and quality of their social relationships [18]. It affects people of all ages, including children and adolescents [19]. In addition to the physical presence of another human being, a person needs relationships that will provide them with a sense of security, trust and belonging. In general, it is assumed that emotional loneliness refers to the absence of an attachment figure (together with feelings of isolation), and social loneliness to the lack of a social network, i.e., the absence of a circle of people that allows an individual to develop a sense of belonging, of company, of being part of a community [20].

Supportive social relationships are an indispensable part of good physical and mental wellbeing. Impaired relations can lead to the development of social loneliness [21]. For many years, authors of studies have pointed to the common prevalence of loneliness among the population [22,23], with higher rates generally reported among adolescents and young children, contrary to the myth that it is more common among older adults [24]. Qualitative data from interviews conducted in a group of 8–10 year-olds have shown that 80% of these children experienced loneliness at school [25].

Loneliness is perceived as a deficit of social relations with regard to an individual’s needs, which is associated with the progression of psychological processes such as depression and anxiety. This association is the strongest during puberty and leads to decreased function in both the physical and mental domains [26,27]. Loneliness is usually the consequence of limited interpersonal contact, while depression is characterized by a decreased mood and lack of satisfaction in all areas of everyday life. In individuals suffering from depressive disorders, loneliness is one of the accompanying symptoms. For this reason, depressive disorders are associated with a sense of loneliness [28].

Mahon et al., Vanhalst et al., and Lasgaard et al. showed that the level of loneliness increases with the severity of depression in adolescents [29,30,31]. However, studies on adolescents are scarce. This study was the first to additionally assess levels of aggression among students.

## 2. Aim of the Study

The aim of the study was to determine the prevalence of anxiety, depression, aggression and sense of loneliness among high school students, and to analyze the correlation between loneliness and depression.

## 3. Materials and Methods

### 3.1. Participants

This was a cross-sectional, descriptive study conducted in the period from October to December 2019 among 300 students of two Polish secondary schools located in one municipality. We used convenience sampling.

Participation in the study was voluntary and anonymous. The questionnaires used in the study, which included an explanation of the aim of the study and instructions on how to complete the forms, were distributed to 320 students during their lessons. The study analysis included 300 correctly completed questionnaires. The study population consisted of 63.0% of students aged 15–16 years and 37.0% of students aged 17–18 years. The mean age of participants was M = 16.63 years; SD = 1.3. Women accounted for 47.3% of the study group and men for 52.7%. In the study group, 34% of the subjects came from urban areas and 66% came from rural areas.

Written consent of the schools’ principals, the students and their parents was obtained for the study. The study was conducted in accordance with ethical principles and was based on informed consent of the participants, who had the right to withdraw from the study at any point without justification. The students’ names were not provided in the questionnaire forms; thus, the data were anonymous.

### 3.2. Instruments

#### 3.2.1. Socio-Demographic Questionnaire

The questionnaire asked about respondents’ sociodemographic data, such as sex, age and place of residence.

#### 3.2.2. Hospital Anxiety and Depression Scale (HADS-M)

In order to evaluate anxiety and depression, the Polish version of the Hospital Anxiety and Depression Scale (HADS–M) adapted by Majkowicz et al. was used [32], which is a modified version of the Hospital Anxiety and Depression Scale (HADS) developed by Zigmond and Snaith [33]. Validation of HADS among adolescents, as performed by White et al., showed that the depression subscale cut-off of 7 gave sensitivity 0.89 and specificity 0.66; an anxiety subscale cut-off of 9 gave sensitivity of 0.83 and specificity 0.47 [34]. In this study, the internal consistency of HADS-M was *a* = 0.65 for anxiety subscale, *a* = 0.71 for depression subscale and *a* = 0.59 for aggression subscale. HADS contains two independent subscales: anxiety and depression; HADS-M includes two additional elements to assess aggression. HADS-M is composed of 16 questions in total; one can score from 0 to 3 points on each of them. The maximum score separately for anxiety (seven questions) and depression (seven questions) is 21 points, and the maximum score for aggression (two questions) is 6 points. A score in the range of 0–7 indicates a lack of disorders, a score in the range of 8–10 means a borderline state, while one in the range of 11–21 indicates the presence of disorders. The following interpretation was adopted in line with the questionnaire key for the anxiety and depression subscales: no disorders: 0–7 points, borderline states: 8–10 points, disorders: 11–21 points; the interpretation for the aggression subscale was: 0–2 points: no disorders, 3 points: borderline states, 4–6 points: disorders.

#### 3.2.3. De Jong Gierveld Loneliness Scale (DJGLS)

In the study, the Polish version of the De Jong Gierveld Loneliness Scale (DJGLS) developed by De Jong Gierveld [35] was used. The Polish version was validated by Grygiel et al. [36,37]; it correlates with the UCLA Loneliness Scale (*r* = 0.82) and Beck Depression Inventory (BDI) (*r* = 0.46), among other measures. The reliability of DJGLS in the Polish adaptation was assessed using the internal consistency and homogeneity analysis, where the Cronbach alpha consistency level was obtained (α = 0.89) [36]. In this study, the internal consistency of DJGLS was α = 0.87 for emotional loneliness, and α = 0.86 for social loneliness.

The scale includes 11 statements: 6 negative and 5 positive, regarding feelings associated with a lack of satisfaction with social contact and a lack of satisfaction with interpersonal relations. In this scale, two subscales were identified: emotional loneliness and social loneliness. Based on the interpretation of scores for the whole scale, the following categories of loneliness were defined: not lonely (0–2 points); moderate loneliness (3–8 points); severe loneliness (9–10 points); very severe loneliness (11 points). On the emotional loneliness subscale, these include: not lonely (0–2 points), moderately emotionally lonely (3–4 points) and severely emotionally lonely (5–6 points). On the social loneliness subscale, these include: not lonely (0–1 point), moderately socially lonely (2–3 points) and severely socially lonely (4–5 points).

### 3.3. Statistical Method

The normality of data distribution was determined using the Shapiro-Wilk test and homogeneity of variance was checked with the Levene’s test. Pearson’s linear correlation coefficient r and the Student’s t-test for independent variables were applied. The significance level of α = 0.05 was adopted; the results were considered statistically significant if the calculated probability met the condition: *p* ≤ 0.05. Calculations were performed using the Statistica 10.0 package by Statsoft Poland (Warsaw, Poland).

## 4. Results

Overall, on HADS-M, 23% of the students were found to have depressive disorders and were found to be 19.3% borderline abnormal. On the anxiety subscale, 24.0% were found to have disorders and 20.7% to be borderline abnormal; on the depression subscale, borderline abnormal was indicated for 13.3% of the subjects. The percentage of students with disorders on the aggression subscale was 46.3%. Detailed results are presented in Table 1.

Our analysis revealed statistically significant differences between means of the total scores on HADS-M (*p* = 0.002), anxiety subscale (*p* < 0.001) and aggression subscale (*p* = 0.001) with regard to the subjects’ gender. Higher mean values were found for women. A statistically insignificant result was obtained for the depression subscale (*p* = 0.476) (Table 2).

Our analysis did not reveal any statistically significant differences between means in the study group with regard to place of residence. This means that there was no evidence indicating that place of residence was a factor affecting the presence of anxiety, depression and aggression in the study group (Table 3).

In total, on DJGLS, very severe loneliness was demonstrated for 6.7% of the students and severe loneliness for 7.7%. On the emotional loneliness subscale, the percentage of students with severe loneliness (16.7%) was slightly lower compared to the social loneliness subscale (22.7%). Detailed results are presented in Table 4.

Statistically significant differences between means with regard to gender were found only for the emotional loneliness subscale (*p* = 0.008). Women scored higher on this subscale. A statistically insignificant result was obtained for the second component, the social loneliness scale (*p* = 0.514) (Table 5).

A statistical analysis did not reveal any statistically significant differences between means on DJGLS in the study group with regard to place of residence (*p* = 0.292) (Table 6).

Our analysis demonstrated statistically significant positive correlations between HADS-M and DJGLS scores. Increasing loneliness scores were accompanied by increasing anxiety, depression and aggression scores. The highest strength of effect was found between two overall indicators: the total HADS-M score and the total loneliness scale score (*r* = 0.61). The r values between the different subscales were diverse, ranging from weak associations (*r* = 0.26) to strong associations (*r* = 0.57). The results are presented in Table 7.

## 5. Discussion

The present study was conducted to determine the prevalence of anxiety, depression, aggression and sense of loneliness among high school students, and to analyze the correlations among loneliness, anxiety and depression. Undoubtedly, attention should be paid to the essence of the problem, which is depression and loneliness among adolescents, and the consequences of delayed or missed diagnosis. For adolescents, monitoring the problem and sufficiently early implementation of support strategies represent a chance to minimize the consequences [38]. For our study, we selected a group of young people who were entering adulthood, were at a stage of many changes and were searching for their own identity. In some adolescents, the associated sense of anxiety and uncertainty as a result of those changes can contribute to the development of mental problems, affecting progression in adulthood, which is consistent with previous research [39,40,41].

The current study demonstrated a higher percentage of adolescents with depressive disorders compared to research by other authors [2,5,7,8]; additionally, it was shown that loneliness was correlated with the frequency of anxiety and depressive disorders [25,28], which were more common in girls [5,6].

The significant correlation between loneliness and the rate of depressive disorders demonstrated in our study is corroborated by the research of other authors [28,42]. In a study by Sahin et al., conducted on high school students, a higher level of loneliness was found in men than women, in contrast to our results. This difference may be due to different living conditions and cultural differences between Turkey and European countries [42].

Shevlin et al. assessed the impact of loneliness on the development of mental disorders in teenagers in Northern Ireland. They found that loneliness, which was present in 15.6% of the subjects, correlated significantly with mental problems and was more common in females. According to the authors, the risk of mental disorders in individuals with a known sense of loneliness increased as much as five-fold [27]. Our study presented similar findings in terms of loneliness scores, and found this variable to be higher in women.

Ebesutani et al. obtained different results regarding the level of loneliness and depression symptoms in a large sample of American adolescents compared to the results of the present study. They demonstrated that 16.8% of adolescents were in an increased score range for depression, 28.7% for loneliness and 9.1% for anxiety. According to the authors of that study, loneliness was a significant predictor of depressive disorders [43]. A study conducted in Norway which aimed to determine a trend in the frequency of symptoms of anxiety, depression and sense of loneliness among adolescents, demonstrated an increase in high depressive symptom loads of up to 73% in a group of girls, and up to 46% in a group of boys. Likewise, the frequency of high anxiety symptom scores increased to 42% in girls and 17% in boys. Severe loneliness was noted in 15% of girls and 9% of boys [44].

Compared to the results of a study by Diehl et al., the current study demonstrated a higher total sense of loneliness score and higher scores on both subscales. Overall, in the study by Diehl et al., moderate loneliness was experienced by 32.4% of the subjects and very severe loneliness by 3.2%. On the emotional loneliness subscale, 7.7% of the subjects were shown to be severely lonely, and on the social loneliness subscale, 3.2% were rated as such. Similar to our study, the authors demonstrated a correlation between loneliness and depression. The differences in scores between the two studies may be due to an age difference. In the study by Diehl et al., the participants were aged 16–29 years. That study was one of the few in which DJGLS was used for the assessment of loneliness [21]. Similar to us, Diehl et al. and the authors of a study on a group of girls aged 15–18 years demonstrated a positive correlation between loneliness and degree of anxiety and depression [45].

Correlation of loneliness on intensity of depression is consistent with the findings of many authors [29,30,31,46], with a significant proportion of studies focusing on the correlation between sense of loneliness and depressive disorders. In a study by Lasgaard et al. conducted on a group of 1009 high school students, a hypothesis was put forward whereby loneliness correlates with the development of symptoms of depression depending on the source of loneliness, and a mutual correlation was demonstrated. Loneliness and depression turned out to be more common in girls, as confirmed in our study. It was demonstrated that peer- and family-related loneliness was associated with depression, anxiety and suicidal thoughts [29]. Similar conclusions were made in a study by Bračič et al. conducted in Slovenia, in which higher loneliness and depressive disorder indicators in adolescents additionally resulted in an increased suicide rate (15.5%) [46]. Vanhalst et al. also pointed to a mutual dependence between loneliness and depression from middle to late puberty [30]. Our findings showed the strength of the association between loneliness and depression to be *r* = 0.61, which was confirmed in a finding from a meta-analysis by Mahon et al., in which the correlation between depression and loneliness also had a high strength of effect [31].

A study by Polish authors Dymowska and Nowicka-Sauer, based on the Primary Care Evaluation of Mental Disorders questionnaire, showed that depressive disorders are present in 25.7% of 18-year-olds [47]. In our study, a similar rate was observed, while the percentage of young people with borderline states was higher. The prevalence of depression symptoms among adolescents was confirmed in a study by Modrzejewska and Bomba [48]. Two groups of adolescents were studied: one in 1984 and the other in 2001. In the first study (1984), depressive disorders were found in 20.8% of boys and 32.2% of girls. In the second study (2001), the Kraków Depression Inventory (KID) was used, which showed that 19.1% of boys and 34.9% of girls had depressive disorders; in addition, the Beck Depression Inventory (BDI) was used, according to which 18.2% of boys and 33.6% of girls had symptoms of depression. Similar to this study, there was a striking finding in a study by Wiklund et al. [49] of a high prevalence of anxiety, both among boys and girls, with a higher prominence in girls. Compared to a group of high school girls in a study by Blom et al. [50], the girls in our study had a higher mean level of anxiety. According to the authors, 31.5% of girls had borderline anxiety and 27% had anxiety disorders compared to 21.1% of boys with borderline anxiety and 8.7% with disorders on the anxiety subscale. Scores on the depression subscale were lower than for anxiety, and gender differences in the level of depression were small and insignificant.

When validating HADS, the original version of HADS-M, on a group of adolescents in Hong Kong, Chan et al. demonstrated the presence of clinically significant anxiety in 30.0% of subjects aged 10–19 years and clinically significant depression in 34% of them. HADS displayed satisfactory psychometric properties as a screening tool for anxiety and depression in a group of adolescents [51].

When comparing the results of the present study with those of other authors, it needs to be emphasized that different research tools were used in the majority of those studies. Out literature review indicated that no existing data that would make a precise comparison possible. Few studies have been conducted using the scales applied in the present study. A small number of students took part in the present study, and convenience sampling was used, which means that only available students were included. Despite this limitation, our study shows that the scale of the problem is substantial. In Poland, few studies have been carried out to assess the level of depression and loneliness among teenagers. These findings showed that loneliness was positively correlated with depression. This study was the first to additionally assess the level of aggression among students. Our study indicates the importance of screening tests for young people.

## 6. Conclusions

In this study, depression and loneliness symptoms were identified in a substantial proportion of the students, and a positive correlation between loneliness and depression was found.

Anxiety and depressive disorders were demonstrated in one in four students. Girls showed higher levels of anxiety, depression and aggression, and higher emotional loneliness.

Early identification of adolescents experiencing loneliness and anxiety and depressive disorders is an important task for parents, teachers and healthcare professionals.

## Figures and Tables

**Table 1 ijerph-18-01706-t001:** Subjects’ HADS-M scores.

HADS-M		*n*	%
Anxiety subscale	No disorders	166	55.3
	Borderline abnormal	62	20.7
	Presence of disorders	72	24.0
Depression subscale	No disorders	236	78.6
	Borderline abnormal	40	13.3
	Presence of disorders	24	8.0
Aggression subscale	No disorders	113	37.7
	Borderline abnormal	48	16.0
	Presence of disorders	139	46.3
Total score	No disorders	173	57.7
	Borderline abnormal	58	19.3
	Presence of disorders	69	23.0

**Table 2 ijerph-18-01706-t002:** HADS-M scores with regard to the subjects’ gender.

HADS-M	Men (*n* = 158)		Women (*n* = 142)		*t*	df	*p*
	Mean	SD	Mean	SD			
Anxiety	6.58	4.28	8.51	4.45	−3.81	298	<0.001
Depression	4.79	3.78	5.11	3.85	−0.71	298	0.476
Aggression	3.01	1.70	3.69	1.74	−3.43	298	0.001
Total score	14.38	8.01	17.30	8.59	−3.05	298	0.002

**Table 3 ijerph-18-01706-t003:** HADS-M scores with regard to place of residence.

HADS-M	Rural (*n* = 198)		Urban (*n* = 102)		*t*	df	*p*
	Mean	SD	Mean	SD			
Anxiety	7.66	4.52	7.18	4.35	0.88	298	0.378
Depression	4.80	3.80	5.22	3.83	−0.90	298	0.369
Aggression	3.42	1.72	3.15	1.81	1.30	298	0.195
Total score	15.88	8.67	15.54	7.90	0.33	298	0.741

**Table 4 ijerph-18-01706-t004:** Subjects’ DJGLS scores.

DJGLS		*n*	%
Emotional loneliness subscale	Not lonely	187	62.5
	Moderate loneliness	63	21.0
	Severe loneliness	50	16.7
Social loneliness subscale	Not lonely	144	48.0
	Moderate loneliness	88	29.3
	Severe loneliness	68	22.7
Total score	Not lonely	130	43.3
	Moderate loneliness	127	42.3
	Severe loneliness	23	7.7
	Very severe loneliness	20	6.7

**Table 5 ijerph-18-01706-t005:** DJGLS scores with regard to gender.

DJGLS	Men (*n* = 158)		Women (*n* = 142)		*t*	df	*p*
	Mean	SD	Mean	SD			
Emotional loneliness	1.77	2.00	2.41	2.14	−2.69	298	0.008
Social loneliness	1.85	1.86	1.99	1.78	−0.65	298	0.514
Total loneliness score	3.61	3.48	4.39	3.58	−1.91	298	0.057

**Table 6 ijerph-18-01706-t006:** DJGLS scores with regard to place of residence.

DJGLS	Rural (*n* = 198)		Urban (*n* = 102)		*t*	df	*p*
	Mean	SD	Mean	SD			
Emotional loneliness	1.99	2.11	2.22	2.05	−0.87	298	0.386
Social loneliness	1.83	1.78	2.07	1.90	−1.06	298	0.290
Total loneliness score	3.83	3.49	4.28	3.63	−1.06	298	0.292

**Table 7 ijerph-18-01706-t007:** Correlations between sense of loneliness and anxiety, depression and aggression.

DJGLS	HADS-M			
	Anxiety	Depression	Aggression	Total Score
Emotional loneliness	*r* = 0.55	*r* = 0.49	*r* = 0.27	*r* = 0.57
	*p* < 0.001	*p* < 0.001	*p* < 0.001	*p* < 0.001
Social loneliness	*r* = 0.47	*r* = 0.52	*r* = 0.26	*r* = 0.54
	*p* < 0.001	*p* < 0.001	*p* < 0.001	*p* < 0.001
Total score	*r* = 0.57	*r* = 0.55	*r* = 0.29	*r* = 0.61
	*p* < 0.001	*p* < 0.001	*p* < 0.001	*p* < 0.001

## Data Availability

The datasets generated and/or analyzed during the current study are not publicly available due to confidentiality, but data is accessible from the corresponding author on reasonable request.

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
