# Peer review of "Loneliness and Depression among Polish High-School Students"

_ijerph, 2021, doi:10.3390/ijerph18041706_

Round 1

Reviewer 1 Report

After reviewing the manuscript entitled “Loneliness and depression among high school students” my opinion is that it could be accepted after major revisions.

The manuscript should not be published if the authors do not address the following comments (especially points 3 and 4):

1) Title.

Rewrite the title to include the study population (high-school Polish students).

2) Abstract.

Correct the typo error: “On the social loneliness subscale, 2 a severe sense of loneliness…

The conclusion included does not reflect the aim of the study (to determine the prevalence of anxiety, depression, aggression and sense of loneliness among high-school students).

3) Introduction.

The authors must include more information about loneliness using references such as those used in the discussion section (e.g., Mahon et al., Vanhalst et al., Lasgaard et al.). Taken into account these previous studies and their results, what research problem is the paper aiming at dealing with? Moreover, what is the novelty of this work?

4) Materials and methods.

The authors need to define the population (size and characteristics) from which the sample was selected.

Why the authors selected 320 (300) participants in this study? How they were selected? Are they representative of the intended population? What is the sampling error? With this type of sampling, the generalizability of results (external validity) is limited.

Questionnaires. Data on reliability and validity are not included. The authors should include information about the psychometric properties of these instruments using studies conducted on adolescents (e.g. White et al.; Grygiel et al.).

How the questionnaires were administered?

How the socio-demographic information was collected?

Statistical procedures. The authors should explain the reason why the Shapiro – Wilk test was used to check the normality of data distribution (n=300).

5) Results.

The authors should reorganize as following order: HADS-M, DJGLS and relationships between HADS-M and DJGLS. Accordingly, tables must be also reordered (and renumbered) as follows:

Actual version

Corrected version

Table 1

Table 4

Table 2

Table 1

Table 3

Table 7

Table 4

Table 2

Table 5

Table 3

Table 6

Table 5

Table 7

Table 6

Data regarding correlations between aggression subscale (HADS-M) and DJGL scores shows weak but significant relationships (r= 0.26 – 0.29; p<0.001). How it could be explained?

6) Discussion.

It seems clear that the use of loneliness and depression screening tools could play a key role to prevent mental health issues in this population. However, and as the authors mentioned, what is the essence of the problem?

This section contains information about loneliness and its relationship with depression in adolescents that should be moved to the introduction.

This study has several significant limitations (e.g., reduced sample size and sampling selection bias) which must be included in the main body of the manuscript (just before the Conclusions section)

7) Conclusions.

The third and fourth conclusions must be removed.

Taken into account the differences observed between males and females, the authors should conclude about this finding.

8) References.

Cite number 47 is not included in the reference list.

Author Response

To: Reviewer 1

Dear Sir/Madam,

We would like to thank the Reviewers for taking the time to read our manuscript, and for their highly valuable comments and suggestions which significantly contributed to improving the quality of the paper.

We hope that the suggested revisions have been made correctly. All revisions in the text of the manuscript are marked in red and by means of the “Track changes” function, with reference provided to the number of the line where a change has been made.

Yours faithfully,

Authors

The text has been edited and supplemented pursuant to the Reviewee’s suggestions.

After reviewing the manuscript entitled “Loneliness and depression among high school students” my opinion is that it could be accepted after major revisions.

The manuscript should not be published if the authors do not address the following comments (especially points 3 and 4):

Comment 1

1) Title.

Rewrite the title to include the study population (high-school Polish students).

Response

The title of the article was changed to Loneliness and depression among high-school Polish students (line1, page 1)

2) Abstract.

Comment 2

Correct the typo error: “On the social loneliness subscale, 2 a severe sense of loneliness…

Response

The figure „2” was removed from the text. It was included in the text by mistake. (line 1, page 27)

“… On the social loneliness subscale, a severe sense of loneliness was found in 22.7% of the students, while on the emotional loneliness subscale, it was found in 16.7% of the subjects”.

Comment 3

The conclusion included does not reflect the aim of the study (to determine the prevalence of anxiety, depression, aggression and sense of loneliness among high-school students).

Response

Conclusions in the Abstract were revised (line 1, page 29-32)

“In this study, a quarter of the students experienced anxiety and depression disorders. Students showing higher levels of anxiety, depression, and aggression also show higher loneliness. Girls show higher levels of anxiety, depression and aggression, as well as emotional loneliness”.

3) Introduction.

Comment 4

The authors must include more information about loneliness using references such as those used in the discussion section (e.g., Mahon et al., Vanhalst et al., Lasgaard et al.). Taken into account these previous studies and their results, what research problem is the paper aiming at dealing with? Moreover, what is the novelty of this work?

Response

In the Introduction, we referred to a study by Mahon et al., Vanhalst et al., and Lasgaard et al., who showed that the level of loneliness increases with the severity of depression in adolescents. This study was the first to additionally assess the level of aggression among students. The numbering of reference citations was changed for the entire article.

(line 3, page 7-9)

“Mahon et al., Vanhalst et al., and Lasgaard et al. showed that the level of loneliness increases with the severity of depression in adolescents [29,30,31]. However, studies among adolescents are scarce. This study was the first to additionally assess the level of aggression among students”.

4) Materials and methods.

Comment 5

The authors need to define the population (size and characteristics) from which the sample was selected.

Why the authors selected 320 (300) participants in this study? How they were selected? Are they representative of the intended population? What is the sampling error? With this type of sampling, the generalizability of results (external validity) is limited.

Response

In order to conduct the study, we contacted principals of high schools located in one municipality. We received positive responses from two schools, which were included in our study. The study was conducted among 300 students. Convenience sampling was used, which means that only available patients were included in the study. With regard to a small number of secondary schools and students attending these schools, the study group may constitute a representative sample for the population of secondary school students in the municipality where this study was conducted.

(line 4, page 3-9)

“This was a cross-sectional, descriptive study conducted in the period from October to December 2019 among 300 students of two Polish secondary schools located in one municipality. We used convenience sampling. Participation in the study was voluntary and anonymous. The questionnaires used in the study, which included an explanation of the aim of the study and instructions on how to complete the forms, were distributed among 320 students during their lessons. The study analysis included 300 correctly completed questionnaires”.

Comment 6

Questionnaires. Data on reliability and validity are not included. The authors should include information about the psychometric properties of these instruments using studies conducted on adolescents (e.g. White et al.; Grygiel et al.).

Added in the Methodology section:

Validation of HADS among adolescents performed by White et al. showed that the depression sub-scale cut-off of 7 gave sensitivity 0.89 and specificity 0.66; anxiety sub-scale cut- off of 9 gave sensitivity of 0.83 and specificity 0.47 [34].

(line 4, page 27-30)

A new reference item added [34]. White D.; Leach C.; Sims R.; Atkinson M. Cottrell D. Validation of the Hospital Anxiety and Depression Scale for use with adolescents. Br. J. Psychiatry 1999, 175: 452–454.

(line 12)

“…was used. The Polish version was validated by Grygiel et al. [36,37], and it correlates with the UCLA Loneliness Scale (r = 0.82) and Beck Depression Inventory (BDI) (r = 0.46), among other measures.

(line 4, page 45-46; line 5, page 1-2)

The reliability of DJGLS in the Polish adaptation was assessed using the internal consistency and homogeneity analysis, where the Cronbach alpha consistency level was obtained (α = 0.89). In this study, the internal consistency of DJGLS was α = 0.87 for emotional loneliness, and α = 0.86 for social loneliness.

Comment 7

How the questionnaires were administered?

Response

The questionnaires with an explanation about the purpose of the study and how to complete it were distributed to students during classes. After completing the questionnaires, the students put them into a prepared box. We obtained consents from the students, parents and the school management.

Comment 8

How the socio-demographic information was collected?

Response

The socio-demographic part of the questionnaire asked about students’ age, sex and place of residence.

We added the following subsections in the manuscript:

3.2 Instruments and 3.2.1 Socio-demographic questionnaire

(line 4, page 19-22)

Comment 9

Statistical procedures. The authors should explain the reason why the Shapiro – Wilk test was used to check the normality of data distribution (n=300).

Response

We used the Shapiro-Wilk test to verify the distribution of normality as it is the most commonly recommended tool in the literature to test the assumption of the normality of the distribution of the analysed feature. It is exhibits higher power than its counterparts (e.g. Kolmogorov-Smirnov test), and it is a test usually used for small samples.

5) Results.

Comment 10

The authors should reorganize as following order: HADS-M, DJGLS and relationships between HADS-M and DJGLS. Accordingly, tables must be also reordered (and renumbered) as follows:

Actual version

Corrected version

Table 1

Table 4

Table 2

Table 1

Table 3

Table 7

Table 4

Table 2

Table 5

Table 3

Table 6

Table 5

Table 7

Table 6

 Response

The order of tables was corrected and the numbers of tables were changed as suggested by the Reviewer.

(line 5-7)

Comment 11

Data regarding correlations between aggression subscale (HADS-M) and DJGL scores shows weak but significant relationships (r= 0.26 – 0.29; p<0.001). How it could be explained?

Response

This means that the increasing values in the total loneliness scale and the emotional loneliness subscale and social loneliness subscale were accompanied by increasing values in the aggression subscale, but the strength of these correlations was low (r = 0.27, r = 0.26, r = 0.29) compared to the correlation between the total HADS-M and DJGLS score, which was moderate (r=0.61).

When characterising the correlation of two features, two factors were presented: direction and strength.

A correlation coefficient (r) ranging from 0 to 1, indicates a positive correlation - it informs that an increase in the value of one feature is accompanied by an increase in the mean values of the other feature.

A correlation coefficient from -1 to 0 indicates a negative correlation - it informs that an increase in the value of one feature is accompanied by a decrease in the mean values of the other feature.

In Table 7, all correlations were positive.

The strength of correlations may be as follows:

below 0.2 - weak correlation

0.2 - 0.4 - low correlation

0.4 - 0.6 - moderate correlation

0.6 - 0.8 - high correlation

0.8 - 0.9 - very high correlation

0.9 - 1.0 - virtually full correlation

In Table 7, the lowest strength of correlation was 0.26, and the highest correlation was 0.61.

6) Discussion.

Comment 12

It seems clear that the use of loneliness and depression screening tools could play a key role to prevent mental health issues in this population. However, and as the authors mentioned, what is the essence of the problem?

This section contains information about loneliness and its relationship with depression in adolescents that should be moved to the introduction.

This study has several significant limitations (e.g., reduced sample size and sampling selection bias) which must be included in the main body of the manuscript (just before the Conclusions section)

Response

We added the following information in the Discussion:

A small number of students (secondary school students) took part in the study, convenience sampling was used, which means that only available students were included in the study.

(line 9, page 54 and line 10 page 1)

We added the following information in the Introduction: „This study was the first to additionally assess the level of aggression among students”.

(line 3, page 9-10)

7) Conclusions.

Comment 13

The third and fourth conclusions must be removed.

Taken into account the differences observed between males and females, the authors should conclude about this finding.

Response

We deleted conclusions 3 and 4

We added conclusion 2

“Anxiety and depressive disorders were demonstrated in one in four students. Girls showed higher levels of anxiety, depression and aggression, and higher emotional loneliness”.

(line 10, page 11-13)

8) References.

Comment 14

Cite number 47 is not included in the reference list.

Response

We added reference position 47, which after adding 1 reference position is now a reference position 51.

  1. Chan, Y.F.; Leung, D.Y.; Fong, D.Y.; Leung, C.M.; Lee, A.M. Psychometric evaluation of the Hospital Anxiety and Depression Scale in a large community sample of adolescents in Hong Kong. Qual Life Res. 2010; 19 (6): 865-73. doi: 10.1007 / s11136-010-9645-1.

(line 12)

Reviewer 2 Report

Thank you for choosing me as a reviewer of this manuscript. The aim of the presented study was to analyze the prevalence of anxiety, depression, aggression, and sense of loneliness among high school students.  320 high school students took part in the study. In my opinion, the authors of the article raise a very important issue regarding depression and sense of loneliness among teenagers. The surveys were conducted in 2019. I must say it would be very interesting to conduct the surveys once again among the same group of high school students now-during the pandemic time and compare the results. This hard time, especially lock-downs (staying at home) must have an impact on mental health. Overall it is a well-written paper.  I would like to congratulate the authors on undertaking the assessment of such important problems.

Author Response

Thank you for choosing me as a reviewer of this manuscript. The aim of the presented study was to analyze the prevalence of anxiety, depression, aggression, and sense of loneliness among high school students.  320 high school students took part in the study. In my opinion, the authors of the article raise a very important issue regarding depression and sense of loneliness among teenagers. The surveys were conducted in 2019. I must say it would be very interesting to conduct the surveys once again among the same group of high school students now-during the pandemic time and compare the results. This hard time, especially lock-downs (staying at home) must have an impact on mental health. Overall it is a well-written paper.  I would like to congratulate the authors on undertaking the assessment of such important problems.

To: Reviewer 2

Dear Sir/Madam,

We would like to thank the Reviewers for taking the time to read our manuscript and for the very valuable opinion and suggestion for continuing research during the Covid-19 pandemic., We are currently compiling the results of a study we have conducted in a much larger population of different age and mental health groups.

We also plan to repeat the survey in the same group of young people, but under the current circumstances.

Yours faithfully,

Authors

Reviewer 3 Report

Thank you for providing me the opportunity to review this manuscript, which reports the findings of a cross-sectional study regarding loneliness and mental health in adolescents. Though the study is not very innovative and does not report many new findings, the topic is certainly in need of research that is germane to the local contexts/countries of authors. Hence, I can see merit in the manuscript. However, there are many unclarities throughout the manuscript that must be addressed. Possibly, some of these unclarities stem from poor English writing. I detail my concerns below.

Abstract:

“… and to analyze the relationship…”.

“Conclusions: A positive relationship…”

It would be more correct to state that the study aimed to analyse the correlation as the study method does not allow determining causal relationships. Any reference to relationships and causality should be rephrased in the abstract and throughout the manuscript.

Abstract:

“… subscale, 2 a severe sense…”. What is the meaning of “2”?

“… early diagnosis of the problem…”: What do you mean with “the problem”. Please be specific.

Introduction

Overall, I have the impression that the introduction can be more clear. The text navigates between mental health problems, mental disorders, anxiety, depression, … There is also a mixture of adolescents, teenagers, young people …

It is also not clear why the introduction addressed stigmatization and the WHO call for intervention. I am not saying that it cannot be relevant. It is not clear why it was included.

In the aims of the study, authors stated that the study also investigated “aggression”. However, this was not introduced in the introduction.

In the data collection there is a difference between social and psychological loneliness. Please address these concepts in the introduction.

What do you mean with latitude as a risk factor? Is there a positive or a negative correlation between latitude and risk? And this is true for mental health problems, total mortality and suicide? How is this relevant for this study?

What do you mean with: “… are affected by severe mental health problems” and “… one in five adolescents are affected by mental health problems”? Have they received a diagnosis?

The studies that are mentioned in the introduction provided different percentages of prevalence of mental health problems. What age groups were included in these studies?

The paragraph starting with: “It is known for a fact that” is not clear. I assume that the issue is that early onset mental health problems may persist into adulthood. However, studies that reported a higher prevalence of mental health problems in (young) adulthood compared to children/adolescents do not provide evidence that early onset disorders can be persistent.

What do you mean with “The right social relations…”? Supportive social relationships?

“… loneliness, which may affect 80% of the population aged 18–65 years [19,20,21], with the highest prevalence among adolescents and children”. What is the prevalence in adolescents and children?

Aim of the study (rather than “the work”)

Please rephrase “the relationship”.

It seems that the study is looking at more correlations than between “loneliness and depression” only.

Materials and Methods

How were the schools selected? What is their socio-economic profile?

What sociodemographic data were collected?

Why did the study collect data about place of residence?

Please provide the Cronbach’s alpha of the HADS-M and DJGLS, and the respective subscales, for this sample.

Can authors use another word for “borderline state” as this might be confused with borderline personality disorder.

In the method section, authors mentioned testing of statistical hypotheses. As this is a cross-sectional study, I would assume that there were no hypotheses. However, if there were hypotheses, they should have been presented before.

Results

Please provide M and SD of sample.

I would include sample description in the Method section, as it is not a result of the analysis.

“The study population consisted of 63.0% of students aged 15–16 years and 37.0% of students aged 17–18 years. Women accounted for 47.3% of the study group and men for 52.7%. In the study group, 34% of the subjects came from urban areas and 66% of the subjects came from rural areas.”

Please present results in the same order as description of instruments in Method section.

What is the meaning of “7” in “The r values 7 between…”?

Discussion

Here, authors introduced a new aim to the study. This should have been addressed in the introduction and aims of the study.

“In the study, we also intended to draw attention to…”.

Throughout the Discussion, any reference to causal relationships between variables must be rephrased.

Just one example, statements such as: “The influence of loneliness on the level of depression …” cannot be made.

In the discussion, authors referred to other studies that have used different methodologies, such as regression analysis. However, it is not because these studies found that a variable had a predictive value, that this is also true for this study.

The discussion should include a paragraph on limitations of the study. For example, potential sampling bias, cross-sectional analysis. Also, data on family context and socio-economic status might have been valuable in the analysis.

Conclusion

Please write in a paragraph, not in dot points.

English language editing is needed. There are several sentences that read a bit awkward. One example: “An analysis by Statistics Poland, the Polish central statistical office (Polish: GUS), of 1996, which was conducted …”.

Please write in active sentences, where possible (e.g., “Statistics Poland analysed … “).

Please be consistent in the use of past or present tense. For example, in the introduction, please refer to studies that have been conducted in the past tense.

Other example: “It is known for a fact that …”

I hope that these few comments may help the authors revising the manuscript. Good luck.

Author Response

To: Reviewer 3

Dear Sir/Madam,

We would like to thank the Reviewers for taking the time to read our manuscript, and for their highly valuable comments and suggestions which significantly contributed to improving the quality of the paper.

We hope that the suggested revisions have been made correctly. All revisions in the text of the manuscript are marked in red and by means of the “Track changes” function, with reference provided to the number of the line where a change has been made.

Yours faithfully,

Authors

The text has been edited and supplemented pursuant to the Reviewee’s suggestions.

Comment 1

Abstract:

“… and to analyze the relationship…”.

“Conclusions: A positive relationship…”

It would be more correct to state that the study aimed to analyse the correlation as the study method does not allow determining causal relationships. Any reference to relationships and causality should be rephrased in the abstract and throughout the manuscript.

 Response

We replaced the „analysis of the relationship” with the „analysis of the correlation” in the entire article.

Abstract:

Comment 2

“… subscale, 2 a severe sense…”. What is the meaning of “2”?

Response

The figure „2” was removed from the text. It was included in the text by mistake.

(line 1, page 27)

Comment 3

“… early diagnosis of the problem…”: What do you mean with “the problem”. Please be specific.

Response

When using the term early diagnosis of the problem, we meant the earliest possible diagnosis of mental problems in adolescents.

The conclusion with this term has been deleted as suggested by the Reviewer 1. We have revised the Conclusion in the Abstract part.

(line 1, page 31-34)

“In this study, a quarter of the students experienced anxiety and depression disorders. Students showing higher levels of anxiety, depression, and aggression also show higher loneliness. Girls show higher levels of anxiety, depression and aggression, as well as emotional loneliness”.

Introduction

Comment 4

Overall, I have the impression that the introduction can be more clear. The text navigates between mental health problems, mental disorders, anxiety, depression, … There is also a mixture of adolescents, teenagers, young people …

Response

We used the terms adolescents and young people interchangeably. According to the WHO definition, the period of adolescence is the period from 10-19 years of age.

The research presented in the Introduction considered this age range, although the selection of age groups differed depending on the authors of the study.

Comment 5

It is also not clear why the introduction addressed stigmatization and the WHO call for intervention. I am not saying that it cannot be relevant. It is not clear why it was included.

 Response

“According to WHO, young people with mental health problems are stigmatized in society and overlooked by healthcare systems [16].”

We referred to the WHO opinion in the Introduction to underline the importance of the problem.

In Poland, the health care system does little to deal with the mental health of children, adolescents and young people.

There diagnostic rates for mental disorders are low in Poland.

“WHO vision of the action plan is a world in which mental health is valued, promoted and protected, mental disorders are prevented and persons affected by these disorders are able to exercise the full range of human rights and to access high quality, culturally-appropriate health and social care in a timely way to promote recovery, in order to attain the highest possible level of health and participate fully in society and at work, free from stigmatization and discrimination” (Mental Health Action Plan, WHO 2013).

Comment 6

In the aims of the study, authors stated that the study also investigated “aggression”. However, this was not introduced in the introduction.

 Response

In this study, the HADS-M scale (16 items) was used, which, compared to the HADS scale (14 items), contains an additional third aggression subscale, which is assessed based on responses to 2 questions.

In our study, we obtained a high score for the aggression subscale, which is very important research information for us.

Comment 7

In the data collection there is a difference between social and psychological loneliness. Please address these concepts in the introduction.

 Response

W added the following text in the Introduction:

Loneliness is a negative feeling that arises when a person perceives their social relationships as quantitatively or qualitatively insufficient [18]. It affects people at people of all ages, including children and adolescents [19]. In addition to the physical presence of another human being, a person needs relationships that will provide them with a sense of security, trust and belonging. In general, it is assumed that emotional loneliness refers to the absence of an attachment figure (together with feelings of isolation) and social loneliness as the lack of a social network, the absence of a circle of people that allows an individual to develop a sense of belonging, of company, of being part of a community [20].

(line 2, page 32-40)    Added References (line 11)

Comment 8

What do you mean with latitude as a risk factor? Is there a positive or a negative correlation between latitude and risk? And this is true for mental health problems, total mortality and suicide? How is this relevant for this study?

 Response

WHO data refer to latitude as countries in a specific geographical region have different characteristics, e.g. social status, living conditions, health protection. These can all be directly related to the number of suicides that can be due to depression.

“An estimated 1.1 million adolescents die each year. The leading causes are road traffic injuries, suicide and interpersonal violence. Millions of adolescents also experience illness and injury. Causes of mortality and morbidity among adolescents differ by sex and age, and also by geographic region” [World Health Organization. Health topics. Available online: https://www.who.int/health-topics/adolescent-health].

Corrected and latitude on by geographic region (line 1, page 41)

Comment 9

What do you mean with: “… are affected by severe mental health problems” and “… one in five adolescents are affected by mental health problems”? Have they received a diagnosis?

 Response

Revised:

Herpertz-Dahlmann et al. showed, that mental disorders include depression, social anxiety, and eating disorders are more common in girls. Their prevalence ranges from 12% to 23%, depending on the particular diagnostic instruments. Disruptive disorders, e.g., disorders of social behavior, are more common in boys, with a worldwide prevalence of approximately 5% to 10% [5].

Bor et al. showed that one in five adolescents experienced mental health problems, again with a higher prevalence among girls [6].

(line 2, page 3-8)

Comment 10

The studies that are mentioned in the introduction provided different percentages of prevalence of mental health problems. What age groups were included in these studies?

Response

According to the WHO definition, the period of adolescence is the period from 10-19 years of age. The research presented in the Introduction considered this age range, although the selection of age groups differed depending on the authors of the study. Data for the European Union and Poland refer to teenagers aged 15-19 years.

Comment 11

 The paragraph starting with: “It is known for a fact that” is not clear. I assume that the issue is that early onset mental health problems may persist into adulthood. However, studies that reported a higher prevalence of mental health problems in (young) adulthood compared to children/adolescents do not provide evidence that early onset disorders can be persistent.

We apologize for the lack of explicitness. Our point was that mental health problems can recur in adulthood.

The sentence was revised as follows: „As indicated in the literature, mental health problems that begin at an early age may recur in adulthood”

(line 2, page 17-18)

Comment 12

What do you mean with “The right social relations…”? Supportive social relationships?

Response

Yes, we meant supportive social relationships. The term has been changed.

(line 2, page 41)

 Comment 13

“… loneliness, which may affect 80% of the population aged 18–65 years [19,20,21], with the highest prevalence among adolescents and children”. What is the prevalence in adolescents and children?

 Response

The data on the incidence of loneliness among children has been presented in more detail.

The text was revised as follows: “For many years, authors of studies have pointed to the common prevalence of loneliness among the population [22,23], with higher rates generally reported among adolescents and young children, contrary to the myth that it is more common among older adults [24]. Qualitative data from interviews conducted in a group of 8-10 year-olds have shown that 80% of these children experienced loneliness at school [25].”

(line 2, page 42-46).

Comment 14

Aim of the study (rather than “the work”) – revised (line 3, page 11)

Please rephrase “the relationship”.

It seems that the study is looking at more correlations than between “loneliness and depression” only.

Response

The aim of the study was harmonised and the term “relationship” was replaced with “correlation”.

(Abstract line 1, page 20; line 3, page 13)

No more correlations were made in the study.

Materials and Methods

Comments 15

How were the schools selected? What is their socio-economic profile?

Response

In order to conduct the study, we contacted principals of high schools located in one municipality. We received positive responses from two schools, which were included in our study. The study was conducted among 300 students. Convenience sampling was used, which means that only available patients were included in the study. With regard to a small number of secondary schools and students attending these schools, the study group may constitute a representative sample for the population of secondary school students in the municipality where this study was conducted.

(line 4, page 3-9)

“This was a cross-sectional, descriptive study conducted in the period from October to December 2019 among 300 students of two Polish secondary schools located in one municipality. We used convenience sampling”. Participation in the study was voluntary and anonymous. The questionnaires used in the study, which included an explanation of the aim of the study and instructions on how to complete the forms, were distributed among 320 students during their lessons. The study analysis included 300 correctly completed questionnaires

Comment 16

What sociodemographic data were collected?

Response

The socio-demographic part of the questionnaire asked about students’ age, sex and place of residence.

We added the following subsections in the manuscript:

3.2 Instruments and 3.2.1 Socio-demographic questionnaire (line 4 , page 19-22).

Comment 17

Why did the study collect data about place of residence?

Response

The schools attended by the students were located in a municipality (in accordance with the administrative division of Poland) which included villages. Students commuted from the countryside to classes in schools located in the city. It was assumed that the incidence of anxiety, depression, aggression and loneliness may differ depending on the place of residence

Comment 18

Please provide the Cronbach’s alpha of the HADS-M and DJGLS, and the respective subscales, for this sample.

Response

We added in the Methods section:

Validation of HADS among adolescents performed by White et al. showed that the depression sub-scale cut-off of 7 gave sensitivity 0.89 and specificity 0.66; anxiety sub-scale cut- off of 9 gave sensitivity of 0.83 and specificity 0.47 [34].

(line 4, page 27-30)

A new reference item added [34]. White D.; Leach C.; Sims R.; Atkinson M. Cottrell D. Validation of the Hospital Anxiety and Depression Scale for use with adolescents. Br. J. Psychiatry 1999, 175: 452–454.

(line 12 )

“…was used. The Polish version was validated by Grygiel et al. [32,33], and it correlates with the UCLA Loneliness Scale (r = 0.82) and Beck Depression Inventory (BDI) (r = 0.46), among other measures. The reliability of DJGLS in the Polish adaptation was assessed using the internal consistency and homogeneity analysis, where the Cronbach alpha consistency level was obtained (α = 0.89) [36]. In this study, the internal consistency of DJGLS was α = 0.87 for emotional loneliness, and α = 0.86 for social loneliness.

(line 4, page 45-46; line 5, page 1-2)

Comment 19

Can authors use another word for “borderline state” as this might be confused with borderline personality disorder.

Response  

Changed according to the original interpretation of the scale on borderline abnormal.

In the interpretation of the original version of the HADS scale was used borderline abnormal.

 (line 1, page 26; line 5, page 22,23,24,30,33,36,39).

Comment 20

In the method section, authors mentioned testing of statistical hypotheses. As this is a cross-sectional study, I would assume that there were no hypotheses. However, if there were hypotheses, they should have been presented before.

Response

No "research hypotheses" were made before in this study. In the Statistical methods secyion, the term referring to "statistical hypotheses" is used rather than research hypotheses. This term has been removed.

(line 5 - Statistical method)

Results

Comment 21

Please provide M and SD of sample.

I would include sample description in the Method section, as it is not a result of the analysis.

“The study population consisted of 63.0% of students aged 15–16 years and 37.0% of students aged 17–18 years. Women accounted for 47.3% of the study group and men for 52.7%. In the study group, 34% of the subjects came from urban areas and 66% of the subjects came from rural areas.”

Response

M and SD were added for the age of respondents, and the socio-demographic characteristics of the group was transferred to the Method section.

(line 4, page 11; line 4, page 20-22)

Comment 22

Please present results in the same order as description of instruments in Method section.

Response

The results were presented as suggested by the Reviewer.

(line 5-7)

Comment 23

What is the meaning of “7” in “The r values 7 between…”?

Response

The figure „7” was removed from the text. It was included in the text by mistake.

“The r values between the different subscales were diverse, ranging from weak associations (r = 0.26) to strong associations (r = 0.57). The results are presented in Table 7.”

(line 7, page 13)

Discussion

Comment 24

Here, authors introduced a new aim to the study. This should have been addressed in the introduction and aims of the study.

“In the study, we also intended to draw attention to…”.

 Response

The statement we used was not the aim of our study and was corrected.

“Undoubtedly, attention should be paid to the essence of the problem, which is depression and loneliness among adolescents, and the consequences of delayed or missed diagnosis.”

(line 8, page 13-15)

Comment 25

Throughout the Discussion, any reference to causal relationships between variables must be rephrased.

Response

The reference to causal relationships was revised in the discussion.(line 8-12)

Comment 26

Just one example, statements such as: “The influence of loneliness on the level of depression …” cannot be made.

In the discussion, authors referred to other studies that have used different methodologies, such as regression analysis. However, it is not because these studies found that a variable had a predictive value, that this is also true for this study.

Response

In the Discussion, we referred to studies which collected data using various research tools.

The sentence was corrected:

“Correlation of loneliness on intensity of depression is consistent with the findings of many authors….”

(line 9, page 10-11)

Comment 27

The discussion should include a paragraph on limitations of the study. For example, potential sampling bias, cross-sectional analysis. Also, data on family context and socio-economic status might have been valuable in the analysis.

Response

We added a paragraph on study limitations:

The main limitations of this study have been identified. This study took place only in two school, thus limiting the generalization of these results to other centers in Poland. A small number of pupils took part in the study, convenience sampling was used, which means that only available pupils were included in the study. W analizach nie uwzględniono kontekstu rodzinnego i statusu socjoekonomicznego. 

(line 10, page 29-32)

Conclusion

Comment 28

Please write in a paragraph, not in dot points.

Response

 As suggested by the Reviewer, we presented the conclusions in paragraphs.

(line 10, page 7-15)

Comment 29

English language editing is needed. There are several sentences that read a bit awkward. One example: “An analysis by Statistics Poland, the Polish central statistical office (Polish: GUS), of 1996, which was conducted …”.

Please write in active sentences, where possible (e.g., “Statistics Poland analysed … “).

Response

The sentences were corrected as suggested by the Reviewer.

(line 2, page 13-14)

“According to the analysis conducted by the Central Statistical Office among 5,407 15-19-year-olds in Poland in 1996, one in ten adolescents experienced mental suffering [8].”

Comment 30

Please be consistent in the use of past or present tense. For example, in the introduction, please refer to studies that have been conducted in the past tense.

Other example: “It is known for a fact that …”

 Response

In the Introduction, the past tense was used for previously conducted research.

Round 2

Reviewer 1 Report

After reviewing the corrected version of the manuscript and the authors' responses document, it can be considered that most of the questions have been corrected in this new version. However, the authors do not clearly address several methodological issues related to sampling procedures (e.g., population definition and sample size error) and how they could affect the external validity of the study.

Author Response

To: Reviewer 1

Dear Sir/Madam,

We thank for taking the time to reread our manuscript and for their next extremely valuable comments.

We hope that the answers provided are correct.

Yours faithfully,

Authors

Comment 1

After reviewing the corrected version of the manuscript and the authors' responses document, it can be considered that most of the questions have been corrected in this new version. However, the authors do not clearly address several methodological issues related to sampling procedures (e.g., population definition and sample size error) and how they could affect the external validity of the study.

Response

According to the demographic data of the Central Statistical Office, the municipality was inhabited by a total of 10,024 people, including 502 adolescents aged 15-19 years (244 women and 258 men) in 2019. The schools in which the study was conducted were attended by 327 students, of which 7 did not participate in the research because they were absent from school on the day of the study, and 20 students did not completely fill the questionnaires.

In our opinion, factors that could have interfered with the external validity of the study included lack of random sample selection (convenient sampling was used) and conducting the study under conditions that are not typical for the majority of society (one urban-rural  municipality). A study conducted in one urban-rural municipality may yield different results than a study conducted in an urban municipality or a large urban agglomeration. The sample size error in this study was 3.5% with a confidence interval of 95%.

Reviewer 3 Report

Dear Authors,

Many thanks for the considerations given to my comments. My only remaining  suggestion would be to include the Cronbach's alpha of the HADS for this sample. 

Good luck with the publication! 

Author Response

To: Reviewer 3

Dear Sir/Madam,

We thank the Reviewers for taking the time to reread our manuscript and for their next extremely valuable comments and suggestions.

We hope that the proposed fixes have been implemented correctly.

All corrections in the text of the manuscript are marked in red and with the use of the "Track changes" function, with reference to the number of the line where the changes were made.

Yours faithfully,

Authors

Comment 1

Many thanks for the considerations given to my comments. My only remaining  suggestion would be to include the Cronbach's alpha of the HADS for this sample. 

Wielkie dzięki za rozważania nad moimi komentarzami. Moją jedyną pozostałą sugestią byłoby włączenie alfa Cronbacha HADS do tej próbki. 

Response

Cronbach’s  alpha of the HADS was included in this sample as suggested by the Reviewer.

“In this study, the internal consistency of HADS-M was a-0,65 for anxiety subscale, a=0,71 for depression subscale and  a=0,59 for aggression subscale”.

(line 4, page 30-31)